# Effect of COVID-19-Induced Changes on Job Insecurity, Presenteeism, and Turnover Intention in the Workplace—An Investigation of Generalized Anxiety Disorder among Hotel Employees Using the GAD-7 Scale

Yeon-Sun Kim [1], Dong-Jin Shin [2,*] and Bo-Kyeong Kim [3,*]

[1] Department of Hotel and Tourism, Wonkwang Health Science University, 514, Iksan-daero, Iksan-si 54538, Jeollabuk-do, Republic of Korea; kysun3113@hanmail.net
[2] Department of Hotel Confectionery and Bakery, Busan Health University, 16, Sari-ro, 55beon-gil, Saha-gu, Busan 49318, Republic of Korea
[3] Department of Tourism Management, Dong-a University, 225, Gudeok-ro, Seo-gu, Busan 49236, Republic of Korea
* Correspondence: sdj8120@naver.com (D.-J.S.); ynnij@hanmail.net (B.-K.K.)

**Abstract:** In this study, we investigated COVID-19′s (coronavirus disease 2019′s) effect on job insecurity, presenteeism, and turnover intention in hotel environments by measuring hotel staffs' generalized anxiety disorder (GAD-7) levels. We surveyed 351 hotel employees from the office, facilities, food and beverage, and cooking departments. Convenience sampling was performed from December 2021 to March 2022. Job insecurity was measured with seven items (easily annoyed, tension, anxiety, nervousness, a lot of worry, fear, uncontrollable worry, restlessness, and discomfort) and demonstrated a significantly positive effect on presenteeism and turnover intention in the high GAD-7 group compared with the low GAD-7 group. Our study contributes academic value to research on GAD-7 in the hotel industry. In addition, it provides a theoretical basis for the relationship between job insecurity and hotel employees' psychological response to the pandemic. Based on the findings, we recommend periodically implementing the GAD-7 scale for employee assessments. Consequently, hotel companies can create guidelines for human resource management post-COVID-19.

**Keywords:** COVID-19; changes in the employment environment; job insecurity; presenteeism; turnover intention; generalized anxiety disorder-7 (GAD-7)

## 1. Introduction

The hotel industry in Korea has suffered heavy losses due to reduced demand from overseas tourists during the COVID-19 pandemic and policy measures such as social distancing [1]. Poor hotel management has caused many employees to leave their jobs. Hotels that could not withstand the business crisis have had to close or reduce their workforce due to labor costs [2]. Hotel companies, which largely provide customer service, have experienced long-term damages. Amidst this rapidly evolving situation, companies have increasingly switched to performance-based pay systems, annual salaries, and wage distribution according to seniority [3]. To overcome the COVID-19 crisis, most hotel companies have sought to reduce the workforce through layoffs, voluntary retirement, and replacing high-paid full-time employees with part-time and contract employees. They have focused on labor cost management by ensuring employee integration, having employees perform multifunctional tasks, etc. [4]. As such, they have promoted changes in the workplace. Owing to high levels of employment insecurity and psychological anxiety, hotel employees' morale has declined, and their turnover intentions have increased [5]. Hotel employees question whether they can maintain their job amid the anxiety associated with changes in their employment environments and relationships due to COVID-19, both of which

have led to job insecurity [6]. In this regard, 48 studies have quantified the prevalence of depressive and anxiety disorders due to COVID-19, reporting 246 million cases of depressive disorder and 374 million cases of anxiety disorder worldwide in 2020 [7]. Therefore, depressive and anxiety disorders increased by 28 and 26%, respectively, compared to pre-pandemic levels. Among the psychological responses to COVID-19, anxiety was the highest at 60.2%, followed by fear at 16.7% and shock at 10.9%. Approximately 50% of people experienced mild or moderate anxiety, and 14% had depression related to COVID-19 [8]. Elsewhere, 21% of Italian adults reported more than severe anxiety, and 7.3% experienced severe insomnia [9]. In addition, 19% of Austrians experienced moderate or severe anxiety due to COVID-19, and 16% experienced severe insomnia [10]. Anxiety was also higher among married people than among single and non-married people.

As such, COVID-19 has not only physical impacts but also psychological and emotional impacts [1]. According to the most recent study on the case of an airline cabin crew, job instability caused by anxiety in the COVID-19 pandemic and inappropriate compensation negatively affected personal achievement [10]. In addition, Lee [11] judged that the relief of employees' depression and anxiety caused by COVID-19 was serious and conducted a study on the need to improve corporate culture marketing to overcome this.

This has resulted in psycho-emotional problems, including generalized anxiety disorder (GAD-7), which affect hotel employees. In this regard, we aimed to measure the GAD-7 of hotel employees due to poor hotel management-induced changes in the employment environment.

The relationship between job insecurity, presenteeism, turnover intention, and job burnout due to changes in the employment environment has also been studied post-COVID-19. However, research on employees' anxiety and psychological states has rarely been conducted, thereby highlighting the novelty of this study. It means that, in unforeseen situations such as COVID-19, the GAD-7 scale of this article is applied to regular hotel workers, not hospital patients.

We examine the effects of hotel environment changes on job insecurity, presenteeism, and turnover intention by dividing hotel employees into two groups: a high GAD-7 group and a low GAD-7 group. Personnel management plans can incorporate our study results as an important aspect of internal marketing. Hotel companies use internal marketing to secure and retain human resources. This approach is crucial to securing both loyal employees and customers. In addition, presenteeism or job loss when an employee's energy is not devoted to the job despite being at work due to COVID-19-induced health problems, negatively affects productivity. Therefore, we consider presenteeism a variable and examine the effects of job instability and presenteeism on employee turnover intention. Based on these findings, we suggest an effective method to increase employees' employment security according to changes occurring in the hotel environment. We also suggest measures to reduce presenteeism and turnover intentions. Finally, we highlight the importance of internal marketing strategies for human resources to increase hotel competitiveness post-COVID-19, as well as the importance of employees' GAD-7.

## 2. Literature Review

### 2.1. Changes in the Employment Environment

Since the COVID-19 outbreak, outsourcing has increased in the domestic hotel industry due to management aggravation. In particular, the existing core departments, such as the front and back offices (management support team), in four-star hotels or lower, have expanded to include indirect employment. In this case, employees are outsourced for housekeeping, parking lot management, beautification, etc., to reduce direct employment [12].

As such, the employment environment has progressed from a supplier-centered environment to a consumer-centered one, breaking the norms associated with a traditional employment relationship [13]. Considering the quantitative changes in hotel companies' employment environments, some of the management strategies that they have applied include outsourcing, layoffs, and voluntary retirement of employees.

In particular, layoffs are the collective termination of employment relationships to promote quantitative flexibility in employment. It is the most common method for companies to improve productivity and rethink management efficiency amid market insecurity and economic crisis risks. Although layoffs affect employees' stress related to job insecurity, they serve as a management method for companies and are a part of their survival strategy in a capitalist economy [7]. In hotel companies, increased uncertainty resulting from organizational reductions and layoffs causes employees' roles to change. These changes directly or indirectly affect their attitudes and behaviors, increasing turnover intentions [14].

Replacing non-regular cooks and part-time workers has been promoted with the slowdown of hotel companies' business growth and the focus on labor cost management. In addition, direct employment in the housekeeping, parking lot management, and beautification departments, which are not hotels' core departments, has been reduced, and indirect employment has increased through outsourcing [12].

Therefore, we regarded the changes in a hotel's employment environment as independent variables. These included both qualitative (e.g., changes considering COVID-19) and quantitative (e.g., changes in the employment environment, including layoffs) organizational changes, as well as increases in non-regular workers.

### 2.2. Job Insecurity

Job insecurity is perceived differently depending on an employee's subjectivity and individual characteristics, recognized as the opposite of job security [15]. Job insecurity first received attention in a study on job stress, which viewed job insecurity as a threat to the continuity of employment relationships. From this viewpoint, Greenhalgh and Rosenblatt [16] defined job insecurity as a sense of helplessness that individuals experience when faced with the threat of losing their job.

Mohr [17] focused on employment sustainability and classified job insecurity as whether individuals can retain their jobs and how certain their future employment status is. Greenhalgh and Rosenblatt [16] presented a model for job insecurity: objective threats to individual influences on individual variables (locus of control, conversation, work orientation, or attribution tendencies) and subjective threats (severity of threat or powerlessness) through intended or unintended organizational messages. As a result, these antecedent variables influence job insecurity, such as increasing propensity level, less effort, and increasing resistance to change. In short, objective information, evaluation, and organizational changes such as mergers, downsizing, and restructuring influence job insecurity and are considered a threat to individuals. In particular, mergers create job insecurity, causing employees to succumb to anxiety and uncertainty [18,19]. In addition, developments in information and communication technologies have emphasized the importance of intellectual skills rather than experience. Therefore, employees who are already accustomed to their existing jobs may experience job insecurity due to this change [20].

### 2.3. Presenteeism

Presenteeism refers to employees' productivity loss while working [21]. Presenteeism is also used to describe a state in which employees appear to be working normally but are actually sick or mentally disengaged [22]. Although scholars have interpreted presenteeism differently, employees' decreasing productivity due to physical and mental health problems integrates these interpretations [23]. Antonio et al. [24] stated that presenteeism negatively affected employees' ability to function, commitment, job satisfaction, organizational health, and productivity. In another study, presenteeism decreased organizational productivity when employees' efficiency was reduced. This loss was estimated to be greater than that caused by absenteeism [25].

Organizational reorganization and coercion to return to work have been found to reduce productivity and cause presenteeism, along with the organizational environment, which includes layoffs and employing non-regular employees to reduce personnel and labor costs after COVID-19 [26]. Presenteeism is influenced by job-related factors, such as

pressure and employment type, as well as individual factors, such as financial situation. In addition, women are more likely to choose presenteeism over absenteeism [27]. Another important factor is job stress. Job stress can cause eating disorders among employees and poor health and can contribute to tardiness and absenteeism. Thus, presenteeism negatively affects a company's productivity [28]. Considering the negative effects of presenteeism mentioned in previous studies, we examined how job insecurity affected presenteeism and turnover intentions.

### 2.4. Turnover Intention

A broader definition of job turnover is when employees cross qualification boundaries as members of society. More narrowly defined, it is when employees voluntarily terminate their relationship with an organization by leaving [29]. Although the intention to leave a job may not immediately result in job turnover, the turnover intention can indirectly affect individuals and organizations [30].

Turnover intention is a major variable for predicting employees' negative job attitudes, such as disengagement, complaints, and dissatisfaction. Therefore, studying turnover intention instead of behavior may be more effective [31]. Turnover intention results from negative job attitudes such as stress, job burnout, cognitive dissonance, emotional dissonance, and presenteeism. It depends on individual perception and judgment [4]. In the case of hotel companies, customer service is crucial because these companies sell services to customers directly. Loyal customers often form a close relationship with employees, so employee turnover may cause a loss of loyal customers. Hence, turnover adversely affects companies' service quality by creating a shortage of skilled workers at the organizational level and issues with workforce management, as well as increasing the cost of recruiting and training new workers. Employee turnover can also create an environment that promotes turnover intention among other employees. Turnover may also negatively impact employees' efforts toward searching for a new job and induce stress. Ultimately, it can exacerbate job transfer stress and personal and interpersonal loss. From the perspective of job security, minimizing the turnover rate can be an efficient management strategy for companies. Therefore, studies on turnover intention are necessary [32].

Previous studies on the turnover intention of hotel employees recognized job environment, insecurity, and stress as variables affecting turnover intention. Studies on presenteeism and turnover intention focusing on employees' GAD-7 status during the COVID-19 pandemic have rarely been conducted. Therefore, a study in this regard would have both academic and practical value.

### 2.5. Generalized Anxiety Disorder-7 (GAD-7)

GAD-7 is a physical symptom arising from tension caused by worry and anxiety. The lifetime prevalence rate of generalized anxiety disorder is 3.7%, higher in countries with high national incomes. Most GAD-7 research is in the medical field, measuring GAD-7 among patients [33]. GAD-7 has also worsened globally since COVID-19. The GAD-7 scale is recognized worldwide as a tool for measuring GAD-7. One of its major advantages is that it identifies anxiety disorders quickly. Therefore, it is widely used in primary healthcare institutions. In this study, we used the GAD-7 scale to measure hotel employees' anxiety during the COVID-19 pandemic. We determined the relationships between GAD and different variables analyzed in this study. In this regard, Korea has not developed a tool for measuring anxiety disorders due to COVID-19. Nonetheless, Jang's study [8], conducted in Korea, can be considered representative. The relationship between anxiety and sleep during the COVID-19 pandemic revealed the significant impact of anxiety levels on sleep quality, insomnia, and total sleep time.

In the present study, we divided hotel employees into high and low GAD-7 clusters based on the seven measurement items of GAD-7. We examined the differences among job instability, presenteeism, and turnover intention according to changes in the employment environment.

## 3. Research Model and Methodology

### 3.1. Research Model and Hypothesis Development

We examined the effects of job insecurity due to changes in a hotel's employment environment, such as presenteeism and turnover intentions, during the COVID-19 pandemic. We investigated hotel employees' turnover intentions, job insecurity, and presenteeism based on GAD-7 measurements. To this end, a research model was established based on previous studies, as shown in Figure 1.

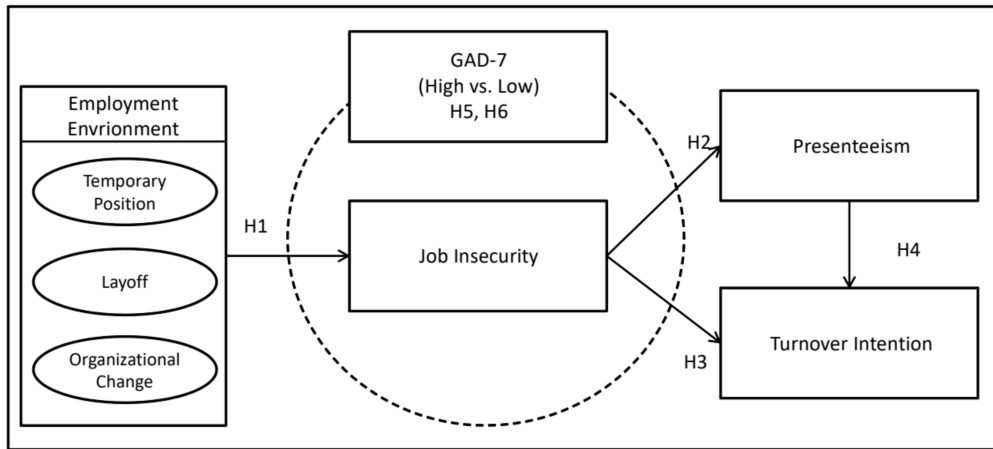

**Figure 1.** Research model.

Hotel companies' changing employment environments include layoffs and cause employees to feel anxious about their jobs [24]. Won and Tak [14] stated that mergers, downsizing, and employment adjustment might cause organizational changes and increase employees' job stress. Even in similar hospitality industries, such as the aviation sector, changes in the employment environment during the COVID-19 pandemic included uncertainty about the future and prolonged unpaid leave, which increased job instability [6]. Previous studies have reported the effects of stress and changes in the workplace (job characteristics and layoffs) on hotel employees during the COVID-19 pandemic [1,8]. They have found that job changes negatively affect job stability and stress, but layoffs have no effect. In response to unexpected changes in the external environment, such as COVID-19, hotel companies have attempted reorganization by introducing changes to the workplace. Such reorganization may eventually affect hotel employees' sense of job insecurity. The following hypotheses were formulated based on previous studies:

**H1:** *Changes in the employment environment will positively affect job insecurity;*

**H1a:** *Temporary workers will positively affect job instability;*

**H1b:** *Layoffs will positively affect job instability;*

**H1c:** *Organizational changes will positively affect job instability.*

Several studies have been conducted on job insecurity and turnover intention, but studies on their relationship with presenteeism are scarce. A recent study showed that hotel employees' job insecurity during COVID-19 significantly affected employee commitment and turnover intention [34,35]. Karatepe et al. [36] suggested that job anxiety increased the likelihood of arriving late for work or leaving early. Arnold and Feldman [37] and Ashford et al. [38] also stated that job insecurity could cause increased turnover intentions among employees.

According to Staufenbiel and König [39], job insecurity had a negative rather than positive effect and, when related to stress, it increased employee turnover intention. Stiglbauer et al. [40] acknowledged that job insecurity increased employee turnover intention and noted the importance of managing stress related to job insecurity. Mauno et al. [41]

stated that job insecurity and turnover intention were closely related and that reducing job insecurity could reduce turnover intention. Other studies have argued that efforts to reduce perceived job insecurity are necessary to prevent the loss of skilled employees [34]. Furthermore, Choi and Jeong [22] suggested that the main cause of presenteeism was the work environment, as it could impact employees' physical and mental health. When job stress and health problems such as depression are prevalent, presenteeism is likely to disrupt corporate productivity and job immersion [42]. Ultimately, presenteeism is associated with physical fatigue, which negatively affects employees' values and sociality [28,43].

Therefore, we assumed that presenteeism affected turnover intention. Based on previous research, the following research hypotheses were formulated:

**H2:** *Job insecurity will positively affect presenteeism;*

**H3:** *Job insecurity will positively affect turnover intention;*

**H4:** *Presenteeism will positively affect turnover intention.*

In this study, the GAD-7 scale was used to examine the psychological behavior of hotel employees and determine the relationships among changes in the workplace, job instability, presenteeism, and turnover intention. Jeong and Hong [44] studied the general public's anxiety about COVID-19 and infection prevention practices. They used the Korean version of the GAD-7 scale to measure COVID-19 anxiety levels, knowledge of COVID-19, possibility of infection, infection prevention, and infection prevention performance. They found statistical differences according to gender, age, marital status, and the type of family living together. The anxiety score was statistically lower for those living together. Reza et al. [45] studied the reliability and validity of the GAD-7 scale for infertile patients. They determined GAD-7 to be a reliable tool for measuring social anxiety disorder in various populations. Another study determined that the tool was concise and easy to use. They considered it a potentially useful tool for identifying anxious psychological states for clinical and research purposes [46]. In this study, Hypotheses 5 and 6 were formulated based on previous studies. The GAD-7 scale was used to determine hotel employees' GAD-7 scores during COVID-19.

**H5:** *Changes in the workplace of the high GAD-7 group will positively affect job insecurity;*

**H5a:** *Temporary workers will positively affect job insecurity;*

**H5b:** *Layoffs will positively affect job insecurity;*

**H5c:** *Organizational changes will positively affect job insecurity;*

**H5d:** *Job insecurity will positively affect presenteeism;*

**H5e:** *Job insecurity will positively affect turnover intention;*

**H5f:** *Presenteeism will positively affect turnover intention;*

**H6:** *Changes in the workplace of the low GAD-7 groups will positively affect job insecurity;*

**H6a:** *Temporary workers will positively affect job insecurity;*

**H6b:** *Layoffs will positively affect job insecurity;*

**H6c:** *Organizational changes will positively affect job insecurity;*

**H6d:** *Job insecurity in the high GAD-7 group will positively affect presenteeism;*

**H6e:** *Job insecurity in the high GAD-7 group will positively affect turnover intention;*

**H6f:** *Presenteeism in the high GAD-7 group will positively affect turnover intention.*

### 3.2. Operational Definition of Variables and the Questionnaire

3.2.1. Changes in the Employment Environment

In this study, hotel survival strategies during COVID-19 included introducing non-regular workers, layoffs, and other organizational changes [6,47]. Previous studies on changes in the employment environment were considered [48]. We revised and supplemented the sub-factors with five questions about non-regular workers, four about layoffs, and three about organizational changes to fit the questionnaire distributed to hotel employees. The questions were measured using a 7-point Likert scale (1 = strongly disagree and 7 = strongly agree).

3.2.2. Job Insecurity

Job insecurity was defined as a sense of helplessness and anxiety among hotel employees about losing or maintaining their jobs based on performance [16,36,48]. We prepared a questionnaire on job insecurity by modifying and supplementing the items developed by Ashford et al. [38] regarding job loss, anxiety, and a sense of helplessness to assess hotel employees. The job insecurity questionnaire included five items: reduced authority within the organization, insecurity regarding control, decline in work value, transfer to another department, and anxiety about workplace cooperation. These items were measured using a 7-point Likert scale (1 = strongly disagree and 7 = strongly agree).

3.2.3. Presenteeism

Presenteeism was defined as employees who could not perform their jobs due to feelings of helplessness and anxiety concerning job insecurity [23,49]. The questionnaire on presenteeism comprised four items: difficulty controlling work stress due to health problems, difficulty completing work, concentration difficulties, and fatigue [6]. The items were measured using a 7-point Likert scale (1 = strongly disagree and 7 = strongly agree).

3.2.4. Turnover Intention

Turnover intention was defined as an employee's intention to leave one company and join another voluntarily [29,50]. The questionnaire included four items: the desire to work for another company, the desire to quit their current company, thoughts about switching to another company, and job searching. These items were measured using a 7-point Likert scale (1 = strongly disagree and 7 = strongly agree).

3.2.5. Generalized Anxiety Disorder (GAD-7)

The questionnaire on GAD-7 included six items: feeling tense, anxious, or agitated; unable to stop worrying; worrying excessively about other things; feeling uncomfortable; restlessness; easily irritated [51,52]. These items were measured using a 7-point Likert scale (1 = strongly disagree and 7 = strongly agree).

### 3.3. Data Collection and Analysis Method

A convenience-sampling questionnaire survey was conducted for hotel employees (office workers and the engineering, food and beverage, and culinary departments). Due to COVID-19, hotel visits were limited, and employees were away for long periods, so completing the survey was difficult. Therefore, the survey was conducted by coordinating with the head of each department in advance. A total of 370 copies were distributed to luxury hotels in Seoul (L Hotel, W Hotel, and P Hotel), Incheon (P Hotel), and Busan (S Hotel) when the number of confirmed COVID-19 cases surged (from December 2021 to March 2022). The questionnaire was double-checked, excluding insincere questionnaires with omissions. An ex ante research strategy was applied to minimize any potential CMV bias due to the self-reported survey. This strategy involved the way the questionnaire was designed and administered (remedy2) [53]. Respondents should be assured of the anonymity and confidentiality of the study, that there were no right or wrong answers, and that they should answer as honestly as possible [54]. A total of 351 copies were used

as verification data for this study. A nominal scale was used for the demographics of the respondents, and a 7-point Likert scale was used for testing the hypotheses. SPSS 21.0 and AMOS 21.0 were used as analysis tools, and a frequency analysis was conducted for the respondents' general information. Exploratory factor, simple regression, and multiple regression analyses were conducted using SPSS 21.0 and AMOS 21.0 for testing the hypotheses. Seven reliable and valid items were used to assess GAD in hotel employees, based on GAD-7 studies from Beard et al. [46] and Löwe et al. [52]. In addition, seven items were subjected to a descriptive statistical analysis. Based on the average of all the items, we classified groups into high and low levels of generalized anxiety disorder and analyzed the relationships between each variable.

## 4. Results

### 4.1. Profile of the Sample

The results of descriptive statistical analysis on the demographic characteristics of the respondents and the GAD-7 scale of this study are shown in Table 1 and Figure 2 respectively.

**Table 1.** Demographic characteristics of respondents (N = 351).

| | Classification | Frequency | Percentage (%) | | Classification | Frequency | Percentage (%) |
|---|---|---|---|---|---|---|---|
| Gender | Male | 255 | 72.6 | Marital status | Married | 250 | 71.2 |
| | Female | 96 | 27.4 | | Single | 101 | 28.8 |
| Education | High school | 60 | 17.1 | Age (years) | 20 | 31 | 8.8 |
| | University /college | 201 | 57.3 | | 30 | 60 | 17.1 |
| | Graduate school | 90 | 25.6 | | 40 | 80 | 22.8 |
| | | | | | 50 | 180 | 51.3 |
| Position | Rank-and-file worker | 47 | 13.4 | Working career | Under 6 months | 21 | 5.9 |
| | Manager | 70 | 19.9 | | months –under 5 years | 37 | 10.5 |
| | Deputy section chief | 141 | 40.2 | | 5 years–under 10 years | 81 | 22.9 |
| | Over section chief | 93 | 26.5 | | 10 years–under 15 years | 57 | 16.1 |
| | | | | | Over 15 years | 157 | 44.5 |
| Depart. | Office worker | 20 | 5.7 | Monthly income (millions earned) | Under 2 | 20 | 5.7 |
| | Engineering | 18 | 5.1 | | 2.5–under 3 | 32 | 9.1 |
| | F&B | 150 | 42.7 | | 3–under 3.5 | 55 | 15.7 |
| | Culinary | 163 | 46.4 | | 3.5–under 4 | 90 | 25.6 |
| Hotel type | Chain hotel | 200 | 57.0 | | 4–under 4.5 | 124 | 35.3 |
| | Local hotel | 151 | 43.0 | | Over 4.5 | 30 | 8.5 |

The results of the descriptive statistical analysis of the GAD-7 found that 'I am easily annoyed or irritated' was the highest, with a mean of 4.34, and 'I worry too much about other things' was second, with a mean of 4.36. On the other hand, 'I am afraid that something terrible will happen' showed the lowest mean of 3.43.

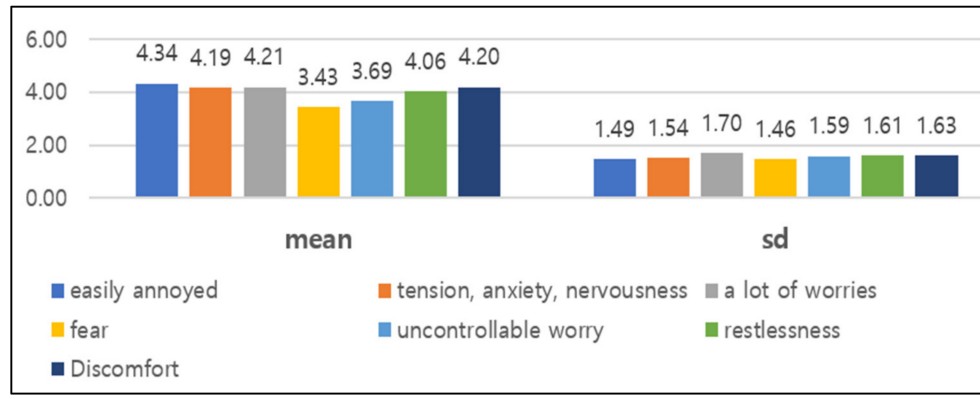

**Figure 2.** GAD-7 descriptive statistics.

### 4.2. Reliability and Validity

### 4.2.1. Changes in the Employment Environment

The reliability and validity verification results of changes in the employment environment are shown in Table 2. The results of the factor analysis found that the commonality of 'my department is highly likely to be integrated' was 0.4 or less, so it was deleted. After deleting it, the commonality of all the factors was above the standard value of 0.6, and the KMO was 0.868, which was over 0.7. The significance probability of the Bartlett sphericity test value was analyzed as $p < 0.000$, which was appropriate. From a factor analysis of changes in employment environment, the three sub-factors of temporary-position workers, layoffs, and organizational changes were extracted, and the total variance explanatory power for the variables was analyzed to be about 77.448%. The Cronbach's alpha value, which indicated internal consistency, was found to be 0.7 or higher. The responses were measured on a 7-point Likert scale (1 = strongly disagree and 7 = strongly agree).

**Table 2.** Results of reliability and validity of changes in the employment environment.

| Construct | Measurement Items | Factor Loading | Eigenvalue | % of Variance | Cronbach's Alpha |
|---|---|---|---|---|---|
| Temporary positions | The number of part-time employees is increasing at the hotel where I work. | 0.900 | 3.904 | 35.492 | 0.888 *** |
| | The number of contract employees is increasing at the hotel where I work. | 0.888 | | | |
| | Some efforts are carried out to reduce manpower at the hotel where I work. | 0.879 | | | |
| | The number of internships is increasing at the hotel where I work. | 0.811 | | | |
| | The number of temporary employees is increasing at the hotel where I work due to COVID-19. | 0.790 | | | |
| Layoffs | Layoffs have already been implemented and will increase in the future at the hotel where I work. | 0.850 | 2.704 | 24.584 | 0.850 *** |
| | Long-term unpaid leave is recommended to reduce labor costs at the hotel where I work. | 0.828 | | | |
| | Involuntary retirement is increasing at the hotel where I work. | 0.681 | | | |
| | Reorganizing will be enforced at the hotel where I work. | 0.645 | | | |
| Organizational Changes | The department I belong to is likely to be downsized. | 0.826 | 1.361 | 12.372 | 0.890 *** |
| | Organizational restructuring is likely to take place in the future. | 0.777 | | | |

KMO: 0.868; Bartlett's test of sphericity: 2395.815; F = 77.448; *** $p < 0.000$.

### 4.2.2. GAD-7 and Job Insecurity

Table 3 shows the reliability and validity of the generalized anxiety disorder (GAD-7) scale and job instability items. The results of the factor analysis showed that the KMO value was 0.874, and the significance probability of the Bartlett sphericity test value was $p < 0.000$, which was analyzed to be suitable as a factor analysis model. Based on these factor analysis results, the items of the GAD-7 scale and job instability were each extracted as a single factor, and the Cronbach's alpha value representing the internal consistency of each factor of the GAD-7 scale and job instability was greater than 0.8. The total variance for the variables was analyzed to be about 60.002%. The responses were measured on a 7-point Likert scale (1 = strongly disagree and 7 = strongly agree).

**Table 3.** Results of reliability and validity of GAD-7 and job insecurity.

| Construct | Measurement Items | Factor Loading | Eigenvalue | % of Variance | Cronbach's Alpha |
|---|---|---|---|---|---|
| GAD-7 | I am easily annoyed and irritated. | 0.823 | 3.613 | 30.105 | 0.879 *** |
| | I feel tense, anxious, or nervous. | 0.745 | | | |
| | I am worried too much about other things. | 0.695 | | | |
| | I fear that something terrible is about to happen | 0.680 | | | |
| | I cannot stop or control worrying. | 0.601 | | | |
| | I am so restless that I find it difficult to sit still. | 0.582 | | | |
| | It is hard for me to be comfortable. | 0.550 | | | |
| Job Insecurity | The authority to take responsibility from start to finish in the performance of work is shrinking. | 0.843 | 3.588 | 29.898 | 0.883 *** |
| | Hotels cannot control what might happen to me. | 0.796 | | | |
| | The value of the importance of my work is likely to be lowered. | 0.793 | | | |
| | There is a possibility of being moved to other tasks of the same job position within the workplace. | 0.582 | | | |
| | There is a possibility of being difficult to cooperate with colleagues. | 0.545 | | | |

KMO: 0.874; Bartlett's test of sphericity: 2308.970; F = 60.002; *** $p < 0.000$.

### 4.2.3. Presenteeism and Turnover Intention

Table 4 shows the reliability and validity verification results of the presenteeism and turnover intention items. The results showed that the KMO was 0.855, which was above the standard value, and the significance probability of the Bartlett sphericity test value was $p < 0.000$, which was suitable as a factor analysis model. The total variance of presenteeism and turnover intention was analyzed to be about 71.739%, and the Cronbach's alpha value representing the internal consistency of each factor of presenteeism and turnover intention was higher than 0.8. The responses were measured on a 7-point Likert scale (1 = strongly disagree and 7 = strongly agree).

**Table 4.** Results of reliability and validity of presenteeism and turnover intention.

| Construct | Measurement Items | Factor Loading | Eigenvalue | % of Variance | Cronbach's Alpha |
|---|---|---|---|---|---|
| Presenteeism | I have a hard time controlling my work stress because of my health problems. | 0.877 | 2.963 | 37.038 | 0.808 *** |
| | I feel that completing my task is too much for me because of my health problems. | 0.819 | | | |
| | I have had difficulty concentrating on my task because of my health problems. | 0.808 | | | |
| | I am tired because I cannot sleep because of my health problems. | 0.764 | | | |

**Table 4.** *Cont.*

| Construct | Measurement Items | Factor Loading | Eigenvalue | % of Variance | Cronbach's Alpha |
|---|---|---|---|---|---|
| Turnover intention | There are times when I want to work for other companies. | 0.909 | 2.776 | 34.701 | 0.901 *** |
| | I am seriously thinking of leaving the company. | 0.819 | | | |
| | I have been thinking about moving to other companies. | 0.715 | | | |
| | I sometimes search for job openings at other companies. | 0.703 | | | |

KMO: 0.855; Bartlett's test of sphericity: 1562.027; F = 71.739; *** *p* < 0.000.

### 4.3. Correlation Analysis

A correlation analysis was performed using the R package install.package ("PerformanceAnalytics") because the density curve and the significance mark appeared on the chart, which was optimally displayed for visualization. The results of the correlation analysis using R to find the linear relationships, namely, linearity between two variables, are shown in Figure 3 and Table 5. All the correlations between the variables in this study showed a positive (+) direction. Specifically, there was a strong correlation between layoffs and temporary-position workers, and job insecurity had strong correlations with GAD-7, presenteeism, and turnover intention.

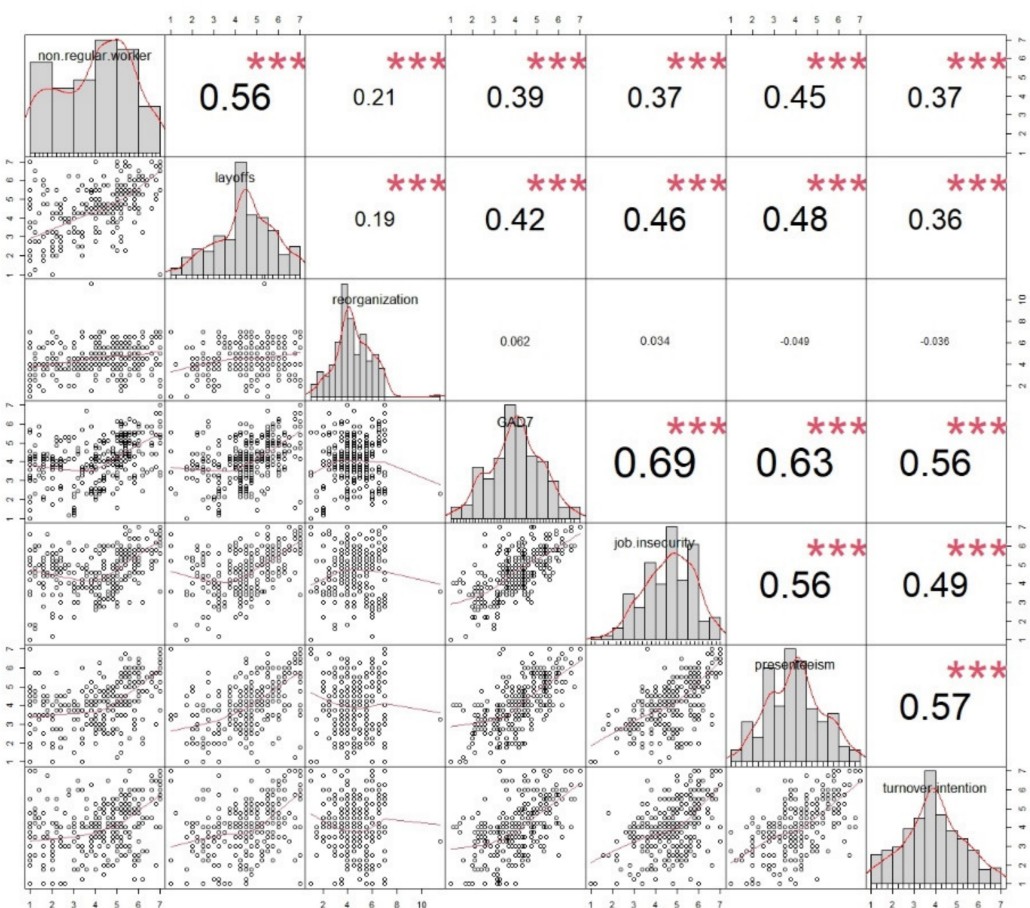

**Figure 3.** Correlation analysis. *** *p* < 0.000.

**Table 5.** Results of correlation analysis.

| | Temporary Positions | Layoffs | Organizational Changes | GAD-7 | Job Insecurity | Presenteeism | Turnover Intention |
|---|---|---|---|---|---|---|---|
| Temporary positions | 1 | | | | | | |
| Layoffs | 0.557 ** | 1 | | | | | |
| Organizational Changes | 0.212 ** | 0.185 ** | 1 | | | | |
| GAD-7 | 0.389 ** | 0.419 ** | 0.062 | 1 | | | |
| Job Insecurity | 0.372 ** | 0.463 ** | 0.034 | 0.687 ** | 1 | | |
| Presenteeism | 0.449 ** | 0.484 ** | −0.049 | 0.634 ** | 0.561 ** | 1 | |
| Turnover intention | 0.366 ** | 0.357 ** | −0.036 | 0.557 ** | 0.489 ** | 0.567 ** | 1 |
| Mean | 4.04 | 4.49 | 4.50 | 4.02 | 4.63 | 3.99 | 3.92 |
| SD | 1.69 | 1.32 | 1.42 | 1.18 | 1.17 | 1.27 | 1.37 |

** $p < 0.005$.

### 4.4. Hypothesis Testing

To verify the hypotheses of this study, the procedure was as follows. First, the sample to achieve the purpose of the study verified the relationships among employment environment change, job insecurity, presenteeism, and turnover intention for 351 people. Second, based on the mean of the GAD-7 scale (M = 4.015), the groups were divided into high and low groups of GAD-7. Then, the relationships between the changes in the employment environment of the two groups and job insecurity, presenteeism, and turnover intention were verified.

4.4.1. Effects of Changes in the Employment Environment on Job Insecurity, Presenteeism, and Turnover Intention

The results of the relationships between job insecurity, presenteeism, and turnover intention and changes in the employment environment for Hypotheses 1, 2, 3, and 4 are shown in Table 6. First, temporary-position workers (t = 3.122) and layoffs (t = 6.665) showed positive (+) effects on job instability. In addition, job insecurity (t = 12.652, t = 10.469) had a positive (+) effect on presenteeism and turnover intention. Finally, presenteeism (t = 12.843) was shown to have a positive (+) effect on turnover intention. From these results, Hypotheses 1, 2, 3, and 4 were adopted, indicating that the influence relationships between all the variables had a positive (+) effect, except for organizational changes on employment environment change.

**Table 6.** Results of the relationships between job insecurity, presenteeism, and turnover intention and changes in the employment environment.

| H | IV | DV | Unstd. Coefficients | | Std. Coefficients | t | *p* | Tolerance | VIF |
|---|---|---|---|---|---|---|---|---|---|
| | | | B | Std. Error | β | | | | |
| 1 | Job Insecurity | Constant | 2.907 | 0.240 | | 12.103 | 0.000 ** | | |
| | | Temporary positions | 0.123 | 0.039 | 0.178 | 3.122 | 0.002 ** | 0.678 | 1.476 |
| | | Layoffs | 0.333 | 0.050 | 0.377 | 6.665 | 0.000 ** | 0.685 | 1.459 |
| | | Organizational changes | −0.060 | 0.040 | −0.073 | −1.522 | 0.129 | 0.948 | 1.054 |
| | | $R^2 = 0.238$, Adj $R^2 = 0.232$, F = 36.165, $p = 0.000$, Durbin–Watson = 1.903 | | | | | | | |
| 2 | Presenteeism | Constant | 1.178 | 0.229 | | 5.140 | 0.000 ** | | |
| | | Job Insecurity | 0.608 | 0.048 | 0.561 | 12.652 | 0.000 ** | 1.000 | 1.000 |
| | | $R^2 = 0.314$, Adj $R^2 = 0.312$, F = 160.085, $p = 0.000$, Durbin–Watson = 1.911 | | | | | | | |
| 3 | Turnover intention | Constant | 1.263 | 0.262 | | 4.827 | 0.000 ** | | |
| | | Job Insecurity | 0.574 | 0.055 | 0.489 | 10.469 | 0.000 ** | 1.000 | 1.000 |

**Table 6.** *Cont.*

| H | IV | DV | Unstd. Coefficients | | Std. Coefficients | t | p | Tolerance | VIF |
|---|---|---|---|---|---|---|---|---|---|
| | | | **B** | **Std. Error** | **β** | | | | |
| | | | R² = 0.239, Adj R² = 0.237, F = 109.608, p = 0.000, Durbin–Watson = 1.652 | | | | | | |
| 4 | Turnover intention | Constant | 1.470 | 0.200 | | 7.350 | 0.000 ** | | |
| | | Presenteeism | 0.614 | 0.048 | 0.567 | 12.843 | 0.000 ** | 1.000 | 1.000 |
| | | | R² = 0.321, Adj R² = 0.319, F = 164.945, p = 0.000, Durbin–Watson = 1.828 | | | | | | |

** $p < 0.001$.

#### 4.4.2. Effect of Changes in the Employment Environment on Job Instability, Presenteeism, and Turnover Intention in the Group with Low GAD-7 (N = 181)

Hypothesis 5 of the low GAD-7 group (N = 181) on the effect relationships of Hypotheses 1, 2, 3, and 4 is shown in Table 7. First, temporary-position workers (t = 1.366) and layoffs (t = 2.465) showed positive (+) effects on job instability. Job insecurity (t = 2.932, t = 3.978) had a positive (+) effect on presenteeism and turnover intention, and presenteeism (t = 5.227) was shown to have a positive (+) effect on turnover intention. Thus, from the results of Hypothesis 5, except for organizational changes in the employment environment, the effect of the group with low GAD-7 (N = 181) among all the variables of positive (+) was adopted.

**Table 7.** Results of the group with low GAD-7 (N = 181).

| H | IV | DV | Unstd. Coefficients | | Std. Coefficients | t | p | Tolerance | VIF |
|---|---|---|---|---|---|---|---|---|---|
| | | | **B** | **Std. Error** | **β** | | | | |
| 5-a | Job Insecurity | Constant | 3.356 | 0.344 | | 9.762 | 0.000 ** | | |
| 5-b | | Temporary positions | 0.081 | 0.060 | 0.114 | 1.366 | 0.174 | 0.756 | 1.324 |
| 5-c | | Layoffs | 0.190 | 0.077 | 0.200 | 2.465 | 0.015 * | 0.793 | 1.260 |
| | | Organizational changes | −0.087 | 0.054 | −0.121 | −1.623 | 0.106 | 0.933 | 1.072 |
| | | | R² = 0.174, Adj R² = 0.159, F = 23.365, p = 0.000, Durbin–Watson = 1.705 | | | | | | |
| 5-d | Presenteeism | Constant | 2.493 | 0.282 | | 8.847 | 0.000 ** | | |
| | | Job Insecurity | 0.199 | 0.068 | 0.214 | 2.932 | 0.004 * | 1.000 | 1.000 |
| | | | R² = 0.214, Adj R² = 0.212, F = 33.085, p = 0.000, Durbin–Watson = 1.784 | | | | | | |
| 5-e | Turnover intention | Constant | 2.050 | 0.321 | | 6.389 | 0.000 ** | | |
| | | Job Insecurity | 0.307 | 0.077 | 0.285 | 3.978 | 0.000 ** | 1.000 | 1.000 |
| | | | R² = 0.139, Adj R² = 0.137, F = 15.828, p = 0.000, Durbin–Watson = 1.758 | | | | | | |
| 5-f | Turnover intention | Constant | 1.892 | 0.278 | | 6.805 | 0.000 ** | | |
| | | Presenteeism | 0.422 | 0.081 | 0.364 | 5.227 | 0.000 ** | 1.000 | 1.000 |
| | | | R² = 0.132, Adj R² = 0.128, F = 27.320, p = 0.000, Durbin–Watson = 1.722 | | | | | | |

* $p < 0.05$; ** $p < 0.001$.

#### 4.4.3. Effect of Changes in the Employment Environment on Job Instability, Presenteeism, and Turnover Intention in the Group with High GAD-7 (N = 170)

Hypothesis 6 of the low GAD-7 group (N = 181) on the effect relationships of Hypotheses 1, 2, 3, and 4 is shown in Table 8. First, temporary-position workers (t = 2.228) and layoffs (t = 3.655) showed positive (+) effects on job instability. Job insecurity (t = 8.857, t = 4.760) had a positive (+) effect on presenteeism and turnover intention as well, and presenteeism (t = 6.432) was shown to have a positive (+) effect on turnover intention. Thus, from the results of Hypothesis 6, except for organizational changes in the employment environment, the effect of the group with high GAD-7 (N = 170) among all the variables of positive (+) was adopted.

**Table 8.** Results of the group with high GAD-7 (N = 170).

| H | IV | DV | Unstd. Coefficients | | Std. Coefficients | t | *p* | Tolerance | VIF |
|---|---|---|---|---|---|---|---|---|---|
| | | | B | Std. Error | β | | | | |
| 6-a | Job Insecurity | Constant | 3.734 | 0.290 | | 12.865 | 0.000 ** | | |
| 6-a | Job Insecurity | Temporary positions | 0.091 | 0.041 | 0.182 | 2.228 | 0.027 * | 0.726 | 1.377 |
| 6-b | | Layoffs | 0.203 | 0.055 | 0.304 | 3.655 | 0.000 ** | 0.703 | 1.422 |
| 6-c | | Organizational changes | 0.023 | 0.047 | 0.035 | 0.486 | 0.628 | 0.942 | 1.061 |
| | | $R^2$ = 0.192, Adj $R^2$ = 0.177, F = 13.139, *p* = 0.000, Durbin–Watson = 1.552 | | | | | | | |
| 6-d | Presenteeism | Constant | 0.954 | 0.433 | | 2.207 | 0.029 * | | |
| 6-d | Presenteeism | Job Insecurity | 0.716 | 0.081 | 0.564 | 8.857 | 0.000 ** | 1.000 | 1.000 |
| | | $R^2$ = 0.318, Adj $R^2$ = 0.314, F = 78.447, *p* = 0.000, Durbin–Watson = 1.546 | | | | | | | |
| 6-e | Turnover intention | Constant | 1.925 | 0.569 | | 3.385 | 0.001 ** | | |
| 6-e | Turnover intention | Job Insecurity | 0.506 | 0.106 | 0.345 | 4.760 | 0.000 ** | 1.000 | 1.000 |
| | | $R^2$ = 0.119, Adj $R^2$ = 0.114, F = 22.659, *p* = 0.000, Durbin–Watson = 1.820 | | | | | | | |
| 6-f | Turnover intention | Constant | 2.163 | 0.388 | | 5.574 | 0.000 ** | | |
| 6-f | Turnover intention | Presenteeism | 0.514 | 0.080 | 0.445 | 6.432 | 0.000 ** | 1.000 | 1.000 |
| | | $R^2$ = 0.198, Adj $R^2$ = 0.193, F = 41.375, *p* = 0.000, Durbin–Watson = 1.632 | | | | | | | |

* $p < 0.05$; ** $p < 0.001$.

## 5. Conclusions

This study was conducted to verify the relationships among job insecurity, presenteeism, and employee turnover intention according to workplace changes in the hotel industry during the COVID-19 pandemic. As we currently face the threat of new mutant viruses after COVID-19, we applied the GAD-7 scale to hotel employees to determine the relationships between different variables and their psychological wellbeing. Specifically, the relationships between the influence of changes in the workplace and job instability, presenteeism, and turnover intention were verified. We achieved the purpose of our study through hypothesis verification. Future research directions are as follows. Firstly, our study had a high percentage of married men working for five-star hotels, and their educational backgrounds were at the college and university level. In terms of age, the proportion of those ≥50 was high. The food and beverage and cooking departments accounted for the highest proportions, and the number of employees with >10 years of work experience was high. The position of assistant manager or higher positions accounted for a large proportion, and their monthly income was over KRW 3.5 million. The study participants were surveyed one year after the onset of COVID-19. We focused on regular employees due to the dismissal of existing contract workers and the closure of business establishments, which represented the reality of hotels.

Secondly, the results of Hypothesis 1 showed that, among the sub-factors of changes in the workplace, non-regular workers and layoffs positively affected job instability, and organizational changes did not affect job instability. These results support the previous research of Yoon and Lee [55], which found that changes in the employment environment due to COVID-19, that is, non-regular workers, organizational changes, layoffs, and outsourcing, had a significant effect on job stress. These results also reflect the current situation of the hotel industry, specifically high turnover rates based on hiring non-regular workers (temporary-position workers) during COVID-19. Middle managers perceive increasing layoffs as job insecurity and a reason for psychological helplessness and anxiety.

Hypotheses 2 and 3 found that job instability positively affected presenteeism and turnover intention, confirming the results of previous research [6,56,57]. In Hypothesis 4, presenteeism had a positive effect on turnover intention. This result supports and confirms a study on fatigue due to health, presenteeism, and turnover intention in a work environment that did not allow rest breaks [20]. This result supports the previous research of Chung [58]

in which presenteeism, one of the job stress factors, had an influence on the job burnout and turnover intention of airline cabin crews. In addition, it means that it is necessary for human resource managers to understand and control job stress to reduce the turnover intention of employees in cases of layoffs and temporary positions.

Finally, the differences between Hypotheses 5 and 6 for the low and high GAD-7 groups' influences on the main variables of this study were as follows. Firstly, regarding the effect of workplace changes on job instability, the group with high GAD-7 for non-regular workers and layoffs had a lower influence than the low GAD-7 group; therefore, the effect on job insecurity was minor. Hotel companies should assess the mental states of their employees by having them conduct a self-diagnosis using the GAD-7 scale at least once a week. Each organization should also keep a checklist. Hotels should prepare remedies or solutions offering psychological stability to employees with high job instability. These solutions include agreements with designated hospitals to relieve hotel employees' anxiety and increase productivity. In addition, periodically implementing the GAD-7 scale indicated that job instability had a stronger influence on presenteeism and turnover intention in the high GAD-7 group than in the low GAD-7 group. Thus, hotels should implement the necessary measures to manage human resources. In the early stages of implementation, work schedules developed according to GAD-7 measurements may be burdensome due to the lack of manpower. However, establishing an efficient human resource management system is competitive for long-term hotel management.

The academic implications of this study are as follows. Firstly, we investigated the relationships among job stress, burnout, and turnover intentions in hotel workplaces during the COVID-19 pandemic, which caused a financial crisis across the hotel industry. In addition, we also measured employees' anxiety disorders according to 'COVID-19 blues' by examining the relationships among job instability, presenteeism, and turnover intentions. This led to not only the theoretical basis for the relationship between employment anxiety and hotel employees' psychological responses, but it applies to the reality of our time practically as well. Although job insecurity has been studied in many fields, this study used the GAD-7 scale for anxiety due to job insecurity caused by external situations, such as COVID-19. We also conducted a study on behaviors corresponding to job insecurity. From this perspective, our study has academic value by combining research related to the hotel industry and psychological factors.

The following are practical implications. In this study, workplace changes negatively affected job insecurity, which increased employees' presenteeism and turnover intentions. Specifically, job instability increased due to layoffs and the employment of non-regular workers. Therefore, to create a stable atmosphere in the workplace and minimize anxiety, human resources managers should consider the aptitude and job competency of their employees. In addition, in unforeseen situations such as COVID-19, the GAD-7 scale can be applied to regular hotel workers, not hospital patients. In other words, lowering hotel employees' anxiety can lower job instability, presenteeism, and turnover intention. Therefore, our study results can provide realistic measures for reducing the turnover rate in the hotel industry, improving employees' work environments, and helping them focus on their work. Despite these implications, this study has a few limitations. Since a GAD-7 scale for COVID-19 has not been developed yet, the study was conducted using the GAD-7 scale and referencing previous studies. Thus, the scale lacked sufficient validity. Therefore, future studies must focus on unexpected situations, such as COVID-19, using the GAD-7 scale. In addition, we conducted our study as a cross-sectional study targeting hotel employees. Continuing progress on cross-sectional studies post-COVID-19 is required in the future.

**Author Contributions:** Conceptualization, Y.-S.K. and D.-J.S.; methodology, D.-J.S. and B.-K.K.; software, D.-J.S. and B.-K.K.; validation, Y.-S.K.; formal analysis, Y.-S.K.; investigation, Y.-S.K.; resources, D.-J.S.; data curation, Y.-S.K.; writing—original draft preparation, Y.-S.K. and D.-J.S.; writing—review and editing, B.-K.K.; visualization, B.-K.K.; supervision, Y.-S.K. and B.-K.K.; project administration, Y.-S.K.; funding acquisition, Y.-S.K. All authors have read and agreed to the published version of the manuscript.

**Funding:** This study was supported by the Wonkwang Health Science University by a 2022 intramural research grant.

**Institutional Review Board Statement:** Ethical review and approval were waived for this study because the Institutional Review Boards of Busan Health University, Dong-a University, and Wonkwang Heath Science University do not require the full review process for consumer acceptance research involving BBI and collecting data exclusively from adults who have adequate decision-making capacity to agree to participate.

**Informed Consent Statement:** Informed consent was obtained from all participants in this study.

**Data Availability Statement:** Data is unavailable due to privacy.

**Acknowledgments:** The authors would like to thank all the supporters and reviewers for giving good advice to improve this study.

**Conflicts of Interest:** The authors declare no conflict of interest.

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
