# Peer review of "Effect of COVID-19-Induced Changes on Job Insecurity, Presenteeism, and Turnover Intention in the Workplace—An Investigation of Generalized Anxiety Disorder among Hotel Employees Using the GAD-7 Scale"

_sustainability, doi:10.3390/su15065377_

Round 1

Reviewer 1 Report

Dear Authors,

The manuscript presents a quite interesting study, with future practical consequences so as to provide  realistic measures to reduce the turnover rate

Must be improved some aspects regarding the arrangement of the paper:

-        Some of the values from the table are not visible. This is the situation from table 2,3,4

-        Eventually the authors can came with future proposals to reduce job instability in this field of activity, in conclusion section.

Author Response

Dear Reviewer 

First of all, thank you for reviewing our article. 

We tried to revise the article very carefully on the basis of your comments  you mentioned as attachment. 

Reviewer 2 Report

Dear Authors,

I recently reviewed your manuscript titled "Effect of changes in the employment environment of hotel employees due to COVID-19 on job insecurity, presenteeism, and turnover intention—focusing on the difference in hotel employees' generalized anxiety disorder (GAD-7) scale" . The study has an interesting subject and draws on a strong theoretical background, and is methodologically well presented. However, I have identified several areas that need improvement in order for the study to reach a higher standard.

The rationale of the work should be given better. The authors need to explain the significance of their study and why it is relevant to the current research.

Literature should be cited more in the introduction. For example, the statement "Significantly, the relationship between job insecurity, presenteeism, turnover intention, and job burnout of hotel employees due to changes in the existing employment environment has been studied even after COVID-19" should be supported by appropriate references.

The sub-dimensions of changes in the employment environment should also be illustrated in the Figure 2 research model.

The font in Figure 3 Descriptive Statistic of GAD-7 is different and needs to be redrawn.

The authors can describe the packages they used when performing the correlation analysis using R.

In the Conclusion section, the authors should provide additional support from the past literature when discussing their findings.

Finally, the authors should use more up-to-date sources in their study.

I believe that these improvements will greatly enhance the quality of your study and bring it to a higher standard. I would like to request a major revision of your manuscript.

Good luck... 

Author Response

Dear Reviewer 

First of all, thank you for reviewing our article. 

We tried to revise the article very carefully on the basis of your comments  you mentioned as attachment. 

Lastly, we used MDPI editing service to check up English language and style. 

Reviewer 3 Report

Dear authors,

To begin with, I am pleased to have had the chance to peruse the article titled, “Effect of changes in the employment environment of hotel employees due to COVID-19 on job insecurity, presenteeism, and turnover intention—focusing on the difference in hotel employees’ generalized anxiety disorder (GAD-7) scale”, which has captured my attention.

The paper has some points that need the author’s attention as below.

1-    The title is very long it’s a paragraph not a title, please follow the scientific rules of writing a research paper title and focus only on your main variable.

2-    In the abstract you mentioned “This study investigated the effect of job instability due to changes in the employment environment of hotel staff during the coronavirus disease 2019 (COVID-19) on job insecurity, presenteeism, and turnover intention by measuring their generalized anxiety disorder (GAD-7)”  please clarify how you measured the impact of job instability on job insecurity ( both constructs are overlapping)

3-    The paper ca benefited from native proofread.

4-     The research novelty and gap as mentioned in your manuscript “However, research on employees' anxiety and psychological state has rarely been conducted, thereby highlighting the novelty of this study” needs more clarification actually there are several studies that tests similar relationships see for example ( please mention how your study is different from those studies)

Elshaer, I.A.; Ghanem, M.; Azazz, A.M.S. An Unethical Organizational Behavior for the Sake of the Family: Perceived Risk of Job Insecurity, Family Motivation and Financial Pressures. Int. J. Environ. Res. Public Health 202219, 6541. https://doi.org/10.3390/ijerph19116541

Elshaer, I.A.; Azazz, A.M.S.; Mahmoud, S.W.; Ghanem, M. Perceived Risk of Job Instability and Unethical Organizational Behaviour Amid the COVID-19 Pandemic: The Role of Family Financial Pressure and Distributive Injustice in the Tourism Industry. Int. J. Environ. Res. Public Health 202219, 2886. https://doi.org/10.3390/ijerph19052886

5-    In the hypothesis’s formulation, no need to mention the sign positive.

6-    Please attach the questionnaire or mention the scale items in table inside the manuscript.

7-    As the study uses self-reported survey, common method variance may be an issue, please clarify how you dealt with CMV

8-    Figures and tables in the manuscript are vague and need to be redesigned

9-    What is this “ <table>

10-  You have some low factor loading, hence I doubt about your discriminant and convergent validity, please do a test for construct validity with composite reliability, average variance extracted and other measure of validity.  

11-  The whole manuscript gives impression that its done without caring, the whole manuscript with figures, table, spaces, spelling need attentions need

12-  The discussion section and implication need more elaboration and comparison with regards to previous similar studies.  

Author Response

(The authors gave the same response as above.)

Round 2

Reviewer 2 Report

The authors successfully made the requested corrections to the paper. As such, I believe the work is at a publishable level. I hope it contributes to the literature.

Reviewer 3 Report

Can Accept in present form